

# Humans read emotional arousal in monkey vocalizations: evidence for evolutionary continuities in communication

Jay W. Schwartz[1,2] and Harold Gouzoules[1]

[1] Department of Psychology, Emory University, Atlanta, GA, United States
[2] Psychological Sciences Department, Western Oregon University, Monmouth, OR, United States

Corresponding author
Jay W. Schwartz, schwartzj@wou.edu

## ABSTRACT

Humans and other mammalian species communicate emotions in ways that reflect evolutionary conservation and continuity, an observation first made by Darwin. One approach to testing this hypothesis has been to assess the capacity to perceive the emotional content of the vocalizations of other species. Using a binary forced choice task, we tested perception of the emotional intensity represented in coos and screams of infant and juvenile female rhesus macaques (*Macaca mulatta*) by 113 human listeners without, and 12 listeners with, experience (as researchers or care technicians) with this species. Each stimulus pair contained one high- and one low-arousal vocalization, as measured at the time of recording by stress hormone levels for coos and the degree of intensity of aggression for screams. For coos as well as screams, both inexperienced and experienced participants accurately identified the high-arousal vocalization at significantly above-chance rates. Experience was associated with significantly greater accuracy with scream stimuli but not coo stimuli, and with a tendency to indicate screams as reflecting greater emotional intensity than coos. Neither measures of empathy, human emotion recognition, nor attitudes toward animal welfare showed any relationship with responses. Participants were sensitive to the fundamental frequency, noisiness, and duration of vocalizations; some of these tendencies likely facilitated accurate perceptions, perhaps due to evolutionary homologies in the physiology of arousal and vocal production between humans and macaques. Overall, our findings support a view of evolutionary continuity in emotional vocal communication. We discuss hypotheses about how distinctive dimensions of human nonverbal communication, like the expansion of scream usage across a range of contexts, might influence perceptions of other species' vocalizations.

## INTRODUCTION

The hypothesis that the ways in which humans express emotions is shared with other species due to common descent dates back to *Darwin (1872)*. Nonverbal vocalizations are a major means of communicating emotion, and the acoustic structures, functions, and

production mechanisms of human nonverbal vocalizations are largely evolutionarily conserved and shared with other mammalian taxa (*Sauter et al., 2010*; *Anikin, Bååth & Persson, 2018*; *Pisanski et al., 2022*). Also, largely conserved across mammals are the acoustic correlates of emotional states within vocalizations (*Briefer, 2012*; *Zimmermann, Leliveld & Schehka, 2013*). For example, emotional arousal (activation of the sympathetic nervous system in association with intense emotions; *Mendl, Burman & Paul, 2010*; *Russell, 2003*) correlates positively with the fundamental frequency (F0; perceived as pitch) of vocalizations across taxa, likely due to tensing of the vocal fold muscles increasing the rate of oscillation (*Scherer, 1986*; *Titze, 1994*; *Riede, 2010*; *Briefer, 2012*). Other acoustic correlates of emotional arousal include F0 modulation, vocal duration, and noisiness (*Briefer, 2012*). The effects of arousal on these acoustic parameters likely reflect evolutionary homologies in the mechanisms of arousal and vocal production.

The hypothesis of evolutionary homology in vocal emotion expression predicts that listeners should be sensitive to the acoustic cues to the emotional states of conspecific and heterospecific vocalizers alike, depending in part on the phylogenetic distance between the two species and the evolutionary history of the particular vocalization in question (*e.g.*, *Fritz et al. (2018)*; though see *Filippi et al. (2017a)*). One approach to testing this prediction is to examine human perceptions of emotion from heterospecific vocalizations (*Linnankoski et al., 1994*; *Nicastro & Owren, 2003*; *Pongrácz et al., 2005*; *Belin et al., 2008*; *McComb et al., 2009*; *Tallet et al., 2010*; *Faragó et al., 2014*; *Scheumann et al., 2014*; *Faragó et al., 2017*; *Maruščáková et al., 2015*; *Filippi et al., 2017a*, *2017b*; *Kelly et al., 2017*; *Fritz et al., 2018*; *Kamiloğlu et al., 2020*; *Merkies, Crouchman & Belliveau, 2021*; *Massenet et al., 2022*). These studies have generally confirmed human sensitivity to emotional information available in heterospecific vocalizations, and thus support *Darwin's (1872)*. hypothesis that significant aspects of human emotional expression are evolutionarily homologous with those of other species.

Listeners have tended to base perceptions of emotional arousal in heterospecific vocalizations predominantly on the mean F0 (*Nicastro & Owren, 2003*; *Pongrácz et al., 2005*; *Faragó et al., 2014*, *2017*; *Maruščáková et al., 2015*; *Filippi et al., 2017a*; *Kelly et al., 2017*), Other significant acoustic parameters have included vocal duration (*Nicastro & Owren, 2003*; *Faragó et al., 2014*, *2017*; *Maruščáková et al., 2015*) and noisiness (*Faragó et al., 2014*; *Filippi et al., 2017b*; *Massenet et al., 2022*). Human listeners appear to show similar perceptual tendencies with regard to the F0 and noisiness of conspecific nonverbal vocalizations (*Juslin & Laukka, 2003*; *Sauter et al., 2010*; *Faragó et al., 2014*; *Schwartz & Gouzoules, 2019*; *Anikin et al., 2021*). Insofar as these acoustic parameters naturally vary according to vocalizer arousal (*Briefer, 2012*), these perceptual tendencies are generally credited with facilitating accurate perceptions of emotion from heterospecific vocalizations (for an exception, see *Kelly et al. (2017)*).

A complicating factor is that many studies in this area have included multiple call types as stimuli (for exceptions, see *Nicastro & Owren (2003)*, *Pongrácz et al. (2005)*, *Filippi et al. (2017a*, *2017b)*, *Faragó et al. (2017)*, *Merkies, Crouchman & Belliveau (2021)* and *Massenet et al. (2022)*), making it difficult to know whether listeners were reacting to acoustic variation *within* a single call type, or *between* call types (for more on this distinction, see

*Fischer, Wadewitz & Hammerschmidt (2016)* and *Schwartz, Engelberg & Gouzoules (2020)*). Distinct call types typically serve disparate social functions and potentially reflect differing cognitive processes, whereas acoustic variation within a call type more reliably relates to the vocalizer's emotional state (*Schwartz, Engelberg & Gouzoules, 2020*). It is therefore pertinent to investigate human perception of emotion from acoustic variation within a call type of another species.

An additional issue is whether heterospecific emotion perception is best explained by evolutionary homology, familiarity, or domestication (*Maigrot, Hillmann & Briefer, 2022*). Most previous studies of human perception have used sounds from domestic animals such as cats, dogs, and livestock (for exceptions, see *Filippi et al. (2017a)*, *Kelly et al. (2017)*, *Fritz et al. (2018)* and *Kamiloğlu et al. (2020)*). These, of course, are species with which many humans are familiar—and indeed, which might have undergone evolutionary changes in emotional expression to improve communication with humans (*Pongrácz, Molnár & Miklósi, 2010*). Some studies have suggested accurate perception does not depend on familiarity (*Pongrácz et al., 2005*; *Filippi et al., 2017a*; *Merkies, Crouchman & Belliveau, 2021*; *Maigrot, Hillmann & Briefer, 2022*), but several have demonstrated an association between experience with a particular species and increased accuracy in recognizing the emotional significance of its vocalizations (*Nicastro & Owren, 2003*; *McComb et al., 2009*; *Tallet et al., 2010*; *Scheumann et al., 2014*; *Faragó et al., 2017*; *Parsons et al., 2019*). This raises the question of the degree to which human accuracy in heterospecific emotion recognition tasks reflects evolutionary homology, as opposed to a capacity to learn to recognize any individual species' unique emotional expressions through exposure.

The present study examined human perceptions of emotional arousal from acoustic variation within call types of rhesus macaques (*Macaca mulatta*), a cercopithecid primate whose lineage is estimated to have diverged from that of humans c. 30 million years ago (*Steiper & Young, 2008*). Although rhesus macaques are common in research institutions and in urban areas in South Asia, many humans have little to no direct experience observing or interacting with them, presenting an opportunity to test the degree to which humans can perform fine-grained discriminations within call types of an undomesticated, unfamiliar species. The rhesus macaque vocal repertoire contains a variety of call types including coos—tonal vocalizations associated with social functions ranging from signaling the presence of food to reuniting separated kin (*Bayart et al., 1990*; *Hauser, 1991*; *Hauser & Marler, 1993*)—and screams—high-F0 vocalizations thought to be evolutionarily homologous with the screams of other primates including humans, usually produced during agonistic interactions to recruit aid from matrilineal kin and allies (*Gouzoules, Gouzoules & Marler, 1984*; *Gouzoules, Gouzoules & Tomaszycki, 1998*; *Gouzoules, 2005*). A recent study that compared stress hormone levels and coo vocalizations produced in a formal behavioral test (Human Intruder Test; *Kalin & Shelton, 1989*), and also compared screams produced during naturally during social interactions that varied in the degree of aggression received by the vocalizer, suggested that both call types exhibit within-type acoustic variation that correlates with the emotional arousal of the vocalizer (*Schwartz, Sanchez & Gouzoules, 2022*). The present study examined whether human listeners are sensitive to the emotional significance of variation among coos and

screams, and assessed the role of experience, other participant characteristics, and acoustic properties.

In a previous study, human listeners did not accurately discriminate between positively and negatively valenced rhesus macaque vocalizations including screams (*Fritz et al., 2018*), although human listeners have shown sensitivity to other kinds of acoustic information in this species' calls (*Fugate, Gouzoules & Nygaard, 2008*). Previous research has also shown sensitivity to arousal cues in the vocalizations the closely related Barbary macaque, *Macaca sylvanus* (*Filippi et al., 2017a*). We therefore hypothesized that participants would accurately perceive differences in emotional arousal within pairs of coos and of screams. We also examined perceived arousal differences between the two call types.

An additional objective was to test the hypothesis that familiarity affects sensitivity to the emotional significance of another species' vocalizations (*Nicastro & Owren, 2003*; *Tallet et al., 2010*; *Parsons et al., 2019*). We also examined the effects of empathy (Cambridge Behaviour Scale; *Baron-Cohen & Wheelwright, 2004*), human emotion recognition (Cambridge Face-Voice Battery; *Golan, Baron-Cohen & Hill, 2006*; *Golan et al., 2007*), and attitudes toward animal welfare (Animal Attitudes Scale; *Herzog, Grayson & McCord, 2015*). The hypothesis that such characteristics might correlate with the ability to recognize emotion from heterospecific vocalizations is intriguing, but previous research to date has provided little support for this notion (*Maruščáková et al., 2015*).

We predicted that mean F0 would play a role in participants' perceptions of arousal, in line with previous research on human emotion perception, and with the positive relationship between arousal and the overall F0 of rhesus macaque coos and screams (*Schwartz, Sanchez & Gouzoules, 2022*). We also examined the role of noisiness and duration of vocalizations, since these have been shown by the studies cited above to affect listeners' perceptions of the emotional states of animals, and F0 range, which appears to correlate with emotional arousal in rhesus macaque coos (*Schwartz, Sanchez & Gouzoules, 2022*).

## MATERIALS AND METHODS

### Ethical statement

All animal procedures were approved by the Emory University Institutional Animal Care and Use Committee (YER-2003417-032317A) in accordance with the Animal Welfare Act and the U.S. Department of Health and Human Services "Guide for Care and Use of Laboratory Animals". Procedures involved in sourcing vocalizations used as stimuli for this study took place as part of longitudinal studies at the Emory National Primate Research Center (ENPRC) funded by the U.S. National Institutes of Health NIH/NICHD R01 grant HD077623 (Principle Investigators: M. M. Sanchez & M. E. Wilson), independent of this study of vocal perception. Human testing procedures were approved by and conducted in compliance with the Emory University Institutional Review Board (IRB00102796).

## Participants

One hundred thirteen undergraduate students at Emory University and volunteers from the Atlanta area comprised our "inexperienced" sample; each indicated no experience working with or observing rhesus macaques. Emory undergraduate participants were recruited *via* an online portal, and received credit in an introductory psychology course; other participants contacted the researchers after seeing fliers advertising the study posted around Emory's campus, and were not compensated in any way. Twelve individuals employed by or affiliated with the ENPRC comprised our "experienced" sample, each reporting a non-zero number of months working with rhesus macaques (mean ± SD = 111 ± 76 months). These participants volunteered after hearing about the study through an email announcement or word-of-mouth from other ENPRC employees. All who were recruited using the above methods were invited to complete the study; all those who did were sampled.

Written informed consent was obtained from all individual participants included in the study. Each participant reported their gender (inexperienced: 73% female, 27% male, 0% other; experienced: 67% female, 33% male, 0% other), age (inexperienced mean ± SD = 20.2 ± 4.2 years; experienced = 34.2 ± 8.2 years), and native language (inexperienced: 67% English (including 6% natively bilingual), 19% Mandarin/Chinese, 4% Spanish, 3% Korean, 9% other languages (<2% each); experienced: 100% English (including one natively bilingual participant)). All but one of the experienced participants indicated professional work with other taxa in addition to rhesus macaques, including chimpanzees and sooty mangabeys (housed at the ENPRC), pigtailed macaques (formerly housed at the ENPRC), other great apes, lemurs, dogs, and cats. Twenty-one inexperienced participants indicated professional work with animals, including dogs, cats, and/or rodents. Participants were screened for hearing impairments, and disclosure of an impairment that might interfere with perception of auditory stimuli was considered an exclusionary criterion.

## Stimuli

The stimuli used in this study comprised pairs of rhesus macaque vocalizations—either two coos, two screams, or one coo and one scream (hereafter, "mixed")—separated by one second of silence. These vocalizations were collected from infant and juvenile female rhesus macaques living in naturalistic social groups within large outdoor enclosures at the ENPRC Field Station in Lawrenceville, GA, using procedures described in detail in *Schwartz, Sanchez & Gouzoules (2022)*. Briefly, coos of 39 macaques were recorded during Human Intruder Tests (*Kalin & Shelton, 1989*) consisting of three 10-min stages—Alone (the macaque was alone in a roomy cage within a testing room), Profile (a technician wearing a lifelike mask sat motionless at a 90-degree angle to the cage), and Stare (the technician sat facing the cage and directing gaze at the macaque but otherwise remained motionless). A Sony DCR-SR85 video camera rig (Tokyo, Japan) (onboard Sony electret condenser microphone) captured video and audio recordings of the tests, including any vocalizations uttered by the macaques. Audio was digitized in a 48-kHz, 32-bit .wav format. Plasma cortisol concentrations were measured from blood samples obtained

immediately before and immediately after each test; the difference between these two measurements served as an estimation of each macaque's physiological stress reaction to the test. Screams of 18 macaques (a subset of the 39 from which coos were obtained) were recorded during naturally occurring social interactions within the macaques' home enclosures, using a Sennheiser ME66 directional microphone with MZW66 windshield (Wennebostel, Germany) and a Marantz PMD671 solid state recorder (Tokyo, Japan). Recordings were digitized as 44.1-kHz, 16- or 24-bit .WAV files. Each scream bout recording was accompanied by a note of the vocalizer identity and whether the vocalizer received agonistic physical contact prior to the scream(s).

Vocalizations were extracted from larger recordings and prepared for use as stimuli using Adobe Audition CC (Adobe Systems, San Jose, CA, USA). 48-kHz audio from the Human Intruder Test coo recordings was down-sampled to 44.1 kHz and dithered from 32- to 16-bit depth, and 24-bit scream recordings were dithered to 16-bit depth. The Human Intruder Testing room contained a noise generator producing brown noise at high volume, to create a controlled and constant auditory background for testing; a noise reduction algorithm was used to prepare the coos for presentation and acoustic analysis (described in detail in *Schwartz, Sanchez & Gouzoules (2022)*). The generally very high F0 of the screams (2–10 kHz), permitted removal of background noise at lower frequencies; this was done by selecting noise at lower frequencies using the marquee and lasso selection tools in Audition, and reducing the sound level of the selection. After noise reduction/ removal, all vocalizations underwent RMS amplitude normalization (following best practices outlined by *Owren & Bachorowski (2007)*) in Audition, using the equal loudness contour (which weights frequency amplitudes according to their perceptual loudness). As a result, most of the vocalizations' sound envelopes occupied approximately −12 dB, while the remaining background noise of the coos occupied a sound level of approximately −42 dB. Final waveforms and spectrograms of individual vocalizations were inspected to ensure that they contained no clipping or other distortions.

To construct the coo stimulus pairs, 6-month-old macaques were paired into dyads consisting of one high-arousal macaque and one low-arousal macaque, defined by a difference in cortisol reactivity (*i.e.*, increase in plasma cortisol concentration from before until after the test) of >1.5 SD. No dyad exhibited a difference in baseline cortisol concentrations of >1 SD. Every dyad meeting these criteria was used to create stimulus pairs, resulting in 28 dyads consisting of 13 individuals. Individual differences rather than arousal levels inevitably accounted for some acoustic differences among coos within stimulus pairs; in principle, acoustic variation due to individual characteristics should be random and should not be expected to have confounded the results. For each individual, three coos were sampled from the Stare stage: the closest to the 2.5-min mark, the closest to the 5-min mark, and the closest to the 7.5-min mark. Then for each macaque dyad, three coo stimulus pairs were constructed: one pairing the two 2.5-min coos, one pairing the two 5-min coos, and one pairing the two 7.5-min coos. This resulted in a total of 84 coo stimulus pairs, each a combination of 39 individual coos. Each coo in each stimulus pair was preceded and followed by a fade-in and fade-out of the background noise lasting 0.3 s.

Each scream stimulus pair consisted of two screams from a single macaque during two separate social interactions—one scream uttered after experiencing agonistic physical contact (high arousal) and one with no contact (low-arousal). To construct these stimuli, tonal screams (*Gouzoules, Gouzoules & Marler, 1984*) were sampled from the eight macaques from whom tonal screams were recorded during at least one interaction with contact and at least one without. We selected one tonal scream from each bout (*i.e.*, multiple screams emitted in response to a single agonistic interaction) on the basis of recording quality (no clipping, uninterrupted by other vocalizations or loud sounds, *e.g.*, a monkey striking a fence or enrichment structure during a display). All possible combinations of one high-arousal scream and one low-arousal scream from the same individual were exploited to construct stimuli, resulting in 39 stimulus pairs representing 33 screams. Numbers of screams per macaque ranged from 2–9, resulting in 1–18 scream stimulus pairs per macaque.

Each mixed stimulus pair included one scream and one coo; no individual vocalization was repeated across these mixed stimuli. Each of the 33 screams was randomly paired with a coo—either from a subset of the coos comprising the coo stimulus pairs, or randomly chosen from a larger corpus—resulting in 33 mixed stimulus pairs. To account for any potential confounding effects of the presence of background noise in the coo recordings *vs* absence thereof in the scream recordings, we added pure noise, sampled from the Human Intruder Test (coo) recordings, to the scream recordings to match the frequency profile and sound level of the coo recordings, and included 0.3-s fade-ins and fade-outs before and after the screams to match the coos.

The order of the two vocalizations comprising each stimulus pair was alternated so that an equal number of stimulus pairs contained a high-arousal vocalization first and a low-arousal vocalization second, and that each individual vocalization appeared first equally often as it appeared second. Where applicable, the one second of silence between the two sound files began at the offset of the noise fade-out of the first vocalization, and ended at the onset of the noise fade-in of the second vocalization. The stimulus pairs were re-inspected by ear to ensure that the RMS-amplitude normalization employed earlier had in fact resulted in equal perceptual loudness. In a small number of cases where one vocalization sounded louder than the other to the researchers, the amplitude of one was adjusted manually in Audition.

## Procedure

Testing took place in the Bioacoustics Laboratory at Emory University (inexperienced participants, and three of 12 experienced participants), or in an office in the research building at the ENPRC Field Station (nine of 12 experienced participants). After providing their informed consent and demographic information, inexperienced participants completed two pen-and-paper surveys: the 40-item Cambridge Behavioral Scale (*Baron-Cohen & Wheelwright, 2004*), which yields a score between 0 and 80 with higher scores indicating greater empathy, and the 10-item brief Animal Attitudes Scale (*Herzog, Grayson & McCord, 2015*), which yields a score between 0 and 50 with higher scores indicating

greater concern for animal welfare. Due to job-related time constraints, experienced participants did not complete these surveys.

Subsequent procedures took place on a Dell Optiplex 755 desktop computer (Round Rock, TX, USA) running E-Prime 2.0 software (Psychology Software Tools, Inc., Pittsburgh, PA, USA). Participants were instructed that they would hear a series of pairs of vocalizations and that their task was to choose, for each pair, the vocalization that they thought reflected a more intense emotion. They were given the examples of calmness or boredom as reflecting low-intensity emotions, and fear or stress as examples of high-intensity emotional situations. We used the terminology of intense emotions rather than "arousal" in order to circumvent participants' potential unfamiliarity with the formal concept of arousal. Arousal has been described as the intensity of an emotional state (*e.g.*, *Briefer (2012)*; though see *Russell (2003)*), and previous research has shown a strong positive correlation between listener ratings of the 'intensity' and 'activation' (*i.e.*, arousal) of vocal stimuli (*Laukka, Juslin & Bresin, 2005*).

Each participant listened to every stimulus pair through headphones (Beyerdynamic DT 770 Pro; Beyerdynamic GmbH & Co., Heilbronn, Germany, or Sony MDR-CD180; Sony Corporation, Tokyo, Japan). On each trial, they indicated whether they perceived the first or second vocalization to be more emotionally intense by pressing a '1' or '2' key on a peripheral serial response box (model 200a; Psychology Software Tools, Inc., Pittsburgh, PA, USA). Each response triggered a two-second pause followed by the onset of the subsequent stimulus pair. The task was broken into four blocks of stimulus pairs: screams ($N = 39$ stimulus pairs), mixed ($N = 33$), one half of the coo stimuli ($N = 42$), and the other half of the coo stimuli ($N = 42$). These four blocks were presented in a random order, as were the stimulus pairs within each block. After each block, participants were offered the option to take a break lasting as long as they needed, ending when the participant pressed a key indicating they were ready to continue.

The subset of inexperienced participants reporting English as a first language then completed the Cambridge Mindreading Face-Voice (CAM) battery (*Golan, Baron-Cohen & Hill, 2006*; *Golan et al., 2007*). This 100-item task was presented on the same computer; in it, participants view or listen to human emotional facial or vocal expressions and, for each, choose one of four emotion labels to best describe the expression. It yields a score between 0 and 100, with higher scores indicating greater recognition of human emotional facial and vocal expressions. The CAM battery requires a high level of English language proficiency, hence we limited testing to participants who reported English as a first language. Experienced participants did not complete the CAM battery, due to their job-related time constraints. Finally, after completing all other procedures, experienced participants provided written answers to qualitative questions about their knowledge of rhesus macaque vocal communication and their thought process during the task.

## Acoustic analyses

Measurements of mean F0, min F0, max F0, mean harmonic-noise ratio (HNR, with low values indicating a noisier call), and duration were obtained in Praat 6.0.29 (*Boersma & Weenink, 2013*). Spectrograms were generated using fast-Fourier transform with a

Gaussian window shape and 2-ms DFT size. Each vocalization was highlighted manually, looking at the spectrogram and waveform and listening to the vocalization, to obtain its duration while excluding any reverberation. F0 measurements were then obtained from the highlighted portion using the Quantify Source command in the GSU Praat Tools package Version 1.9 (*Owren, 2008*). The default settings for this command were used, with the exception that the pitch ceiling was set to 3,000 Hz for coos and 10,000 Hz for screams to account for the F0 range of these vocalizations. The command uses Praat's To Pitch autocorrelation function to estimate a F0 contour, which the user then inspects and can manually correct if necessary (*e.g.*, octave correction, removal of any unvoiced segments) (*Owren, 2008*). F0 range was calculated as the difference between max and min F0. Mean ± SD acoustic parameter values were as follows: **coos:** mean F0 = 995 ± 169 Hz, F0 range = 340 ± 214 Hz, mean HNR = 13.5 ± 4.32 dB, duration = 0.580 ± 0.145 s; screams: mean F0 = 4,879 ± 1,447 Hz, F0 range = 3,533 ± 2,029 Hz, mean HNR = 0.78 ± 3.31 Hz, duration = 0.504 ± 0.211 s. For each stimulus pair, the relative difference in mean F0, F0 range, mean HNR, and duration between the two vocalizations was calculated as the value for the second vocalization minus the value for the first vocalization, divided by the value for the first vocalization; similarly, the relative difference between the high- and low-arousal vocalization was calculated as the value for the high-arousal vocalization minus the value for the low-arousal vocalization, divided by the latter. Three screams were too noisy to estimate the F0 contour, resulting in four scream stimulus pairs missing relative differences on acoustic parameters other than duration. Responses to these stimulus pairs were excluded from the statistical analyses of the effects of acoustics.

## Statistical analyses

All statistical analyses were conducted in R Statistical Environment (*R Core Team, 2018*) and used an alpha of 0.05. To assess the overall accuracy of participants' responses, each participant was attributed one Accuracy Score per stimulus type (coos, screams, mixed), equal to proportion of stimulus pairs to which the participant responded correctly. For mixed stimulus pairs, selecting the scream was arbitrarily treated as "correct" for the purposes of analysis (but not in interpretation, since arousal was measured using different methods for the two call types, thus one call type cannot be considered objectively higher-arousal than the other). We used one-sample t-tests (t.test command in R) to determine whether the two participant groups—experienced and inexperienced— exhibited significantly greater Accuracy Scores than the 50% accuracy predicted by chance, for each of the three stimulus types. These tests were one-tailed for coo stimuli and scream stimuli (to determine whether participant accuracy was greater than predicted by chance), and two-tailed for mixed stimuli (to determine whether participants selected one call type significantly more often than the other).

To determine the effects of experience working professionally with rhesus macaques on responses, we ran three binomial, logit-link generalized linear mixed models (GLMM)— one for each stimulus type. The outcome variable was whether or not an individual response was correct, and the binomial experience variable (whether the participant had professional work experience with rhesus macaques) was entered as a fixed effect.
To account for non-independence of observations among responses from a single participant, or to a single stimulus pair, participant and stimulus pair were entered as crossed random intercept effects. GLMM were likewise used to determine the effects of other participant characteristics on responses; this analysis was limited to the inexperienced participants' responses since the experienced participants did not complete the surveys or the CAM battery. We again ran one GLMM per stimulus type. The outcome variable for each model and random effects structure were the same; fixed effects included Empathy Quotient, Animal Attitudes Score, CAM Score, and whether a participant had worked with any animal species (other than rhesus macaques) in a professional capacity. Finally, four more GLMM were used to assess the effects of acoustic variables on participant responses—one per participant group for coo and scream stimuli (not mixed stimuli). The random effects structure was the same as the other models; the outcome variable was whether the participant pressed "1" or "2", irrespective of which response was correct. Relative differences in mean F0, F0 range, mean HNR, and duration between the two vocalizations in a stimulus pair were entered as fixed effects. To avoid overfitting, nonsignificant fixed effects were removed from each model. All GLMM were fit by maximum likelihood (Laplace Approximation) using the glmer function in the lme4 package in R (*Bates et al., 2015*). Expected correlations of fixed effect coefficients were checked to avoid multicollinearity. Each nonsignificant fixed effect was individually re-entered into the model to assess its level of non-significance (*Barr et al., 2013*) and check against Type II errors.

## RESULTS

One-sample t-tests indicated that Accuracy Scores (percentage correct) of inexperienced as well as experienced participants were significantly higher than predicted by chance (50%) for coo stimuli (inexperienced: mean = 57.8, 95% CI > 56.7, t = 12.98, $p < 0.0001$; experienced: mean = 54.5, 95% CI > 50.8, t = 2.19, $p = 0.025$); the same was true for scream stimuli (inexperienced: mean = 52.2, 95% CI > 50.8, t = 2.53, $p = 0.006$; experienced: mean = 59.4, 95% CI > 55.4, t = 4.26, $p < 0.001$) (Fig. 1). GLMM indicated that experienced participants responded correctly significantly more often than inexperienced participants on scream stimuli (Table 1; Fig. 1). Overall, inexperienced participants did not show a significant tendency to select coos over screams or vice versa in response to mixed stimuli (mean = 49.5, 95% CI [46.1–52.9], t = −0.289, $p = 0.773$), in contrast to experienced participants, who showed a strong bias towards selecting screams as more emotionally intense (mean = 81.8, 95% CI [68.3–95.4], t = 5.16, $p < 0.001$) (Fig. 1). GLMM indicated that this contrast represented a significant difference between inexperienced and experienced participants (Table 1). In a questionnaire administered after the task, every experienced participant indicated knowledge of the names of the vocalizations they had heard, and six reported knowledge of the contexts in which coos and screams occur. six experienced participants reported an explicit strategy of selecting screams over coos in mixed trials, and four of those six indicated that this strategy was because they perceived the agonistic contexts associated with screams to be more emotionally intense than contexts associated with coos. Finally, among inexperienced participants, those who had

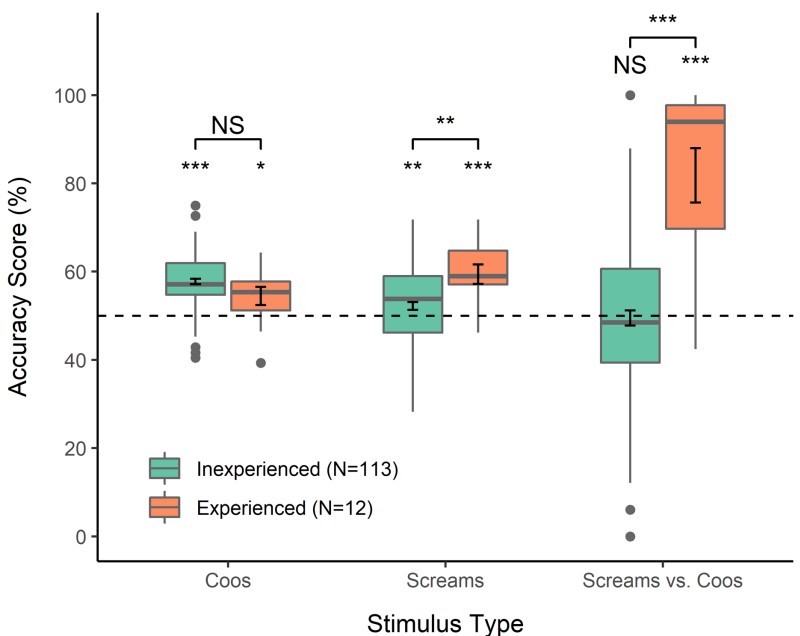

**Figure 1 Accuracy scores.** Calculated for each participant to reflect percent accuracy selecting the vocalization reflecting a more intense emotion, from between two coos, two screams, or one of each. For mixed stimuli, selecting the scream was considered "accurate" for the purposes of analysis and visualization. Error bars within boxes show standard error of the mean. $^*p < 0.05$, $^{**}p < 0.01$, $^{***}p < 0.001$.                                                 

**Table 1 Effects of participant characteristics.**

| Coo stimuli | Est. Coeff. | SE | p value |
|---|---|---|---|
| Experience with rhesus macaques | −0.213 | 0.125 | 0.089 |
| Empathy Quotient | 0.001 | 0.093 | 0.813 |
| Animal Attitudes Score | 0.007 | 0.008 | 0.325 |
| CAM Score | −0.001 | 0.004 | 0.881 |
| Experience with non-primates | −0.226 | 0.093 | 0.015 |
| **Scream stimuli** | | | |
| Experience with rhesus macaques | 0.368 | 0.141 | 0.009 |
| Empathy Quotient | 0.003 | 0.004 | 0.503 |
| Animal Attitudes Score | 0.002 | 0.008 | 0.798 |
| CAM Score | <0.001 | 0.004 | 0.940 |
| Experience with non-primates | 0.004 | 0.110 | 0.968 |
| **Mixed stimuli** | | | |
| Experience with rhesus macaques | 2.428 | 0.379 | <0.001 |
| Empathy Quotient | −0.008 | 0.009 | 0.409 |
| Animal Attitudes Score | −0.012 | 0.018 | 0.514 |
| CAM Score | 0.001 | 0.010 | 0.937 |
| Experience with non-primates | −0.380 | 0.241 | 0.116 |

**Note:**
Model estimates of effects on odds of correct responses or, for mixed stimuli, odds of selecting a scream over a coo. Characteristics other than experience with rhesus macaques were tested only among participants inexperienced with primates.

**Table 2 Descriptives for and effects of acoustic parameters.**

| Coo stimuli | Mean ± SD | Inexperienced | | | Experienced | | |
|---|---|---|---|---|---|---|---|
| | | Est. Coeff. | SE | p value | Est. Coeff. | SE | p value |
| Mean F0 | 0.078 ± 0.274 | 2.818 | 0.480 | <0.001 | 3.316 | 0.534 | <0.001 |
| F0 Range | 0.866 ± 1.728 | 0.149 | 0.074 | 0.043 | 0.126 | 0.094 | 0.184 |
| Mean HNR | −0.152 ± 0.373 | −0.933 | 0.175 | <0.001 | −1.033 | 0.209 | <0.001 |
| Duration | 0.011 ± 0.374 | 1.936 | 0.310 | <0.001 | 2.917 | 0.420 | <0.001 |
| **Scream stimuli** | | | | | | | |
| Mean F0 | 0.282 ± 0.385 | −0.272 | 0.429 | 0.525 | 2.061 | 0.552 | <0.001[a] |
| F0 Range | 0.811 ± 1.438 | 0.315 | 0.106 | 0.003 | 0.056 | 0.372 | 0.710 |
| Mean HNR | −3.013 ± 7.821 | 0.002 | 0.004 | 0.685 | −0.004 | 0.007 | 0.587 |
| Duration | 0.779 ± 0.917 | 0.088 | 0.178 | 0.619 | 1.088 | 0.223 | <0.001[a] |

Notes:
Left column shows relative differences in acoustic parameters between high- and low-arousal vocalizations within stimulus pairs; positive indicates higher-arousal calls had higher mean values. Center and right columns show model estimates of effects of relative change in the acoustic parameter from the first to the second vocalization on the odds of selecting the second vocalization as reflecting a more intense emotion; positive indicates the call with the higher value was more likely to be selected (regardless of whether doing so was correct).
[a] Effects estimated individually due to collinearity.

professional work experience with a non-primate animal responded significantly less accurately to coo stimuli than those who had not worked with any animals (Table 1); we did not find any other significant effects of participant characteristics on responses.

On coo trials, inexperienced and experienced participants alike most often selected as more emotionally intense the coo that was higher in mean F0, lower in HNR (*i.e.*, noisier), or longer in duration, while inexperienced participants also showed a slight tendency to select the coo showing a wider F0 range (Table 2; Fig. 2). Congruent with this, the higher-arousal coo in a pair was, on average, higher in mean F0 and F0 range, and lower in HNR, though some of these average relative differences were minute and relative differences among all parameters ranged from negative to positive (Table 2). Plotting the coo pairs by relative mean F0 difference and average participant response (Fig. 2A) shows that, for most (though by no means all) of the pairs in which the second coo represented greater arousal than the first (orange/light points), the second coo was also higher in F0 (*i.e.*, relative F0 difference >0), and was more often selected by participants (*i.e.*, proportion of '2' responses >0.5). A strong positive correlation is also visible between relative duration difference and proportion of '2' responses, but this does not appear to have translated into correct responses; if anything, it seems that in the majority of coo pairs, the coo representing greater arousal was shorter in duration (*i.e.*, more orange/light points falling at relative duration difference <0) (Fig. 2A), though the average duration difference between high- and low-arousal coos was close to zero (Table 2).

On scream trials, inexperienced participants showed a slight tendency to select the scream exhibiting a wider F0 range, while experienced participants tended to select the scream exhibiting a higher mean F0 and longer duration (Table 2; Fig. 2B). Including mean F0 or duration individually in the experienced participant GLMM showed significant effects of each (Table 2), however, including both predictors in a single model caused one

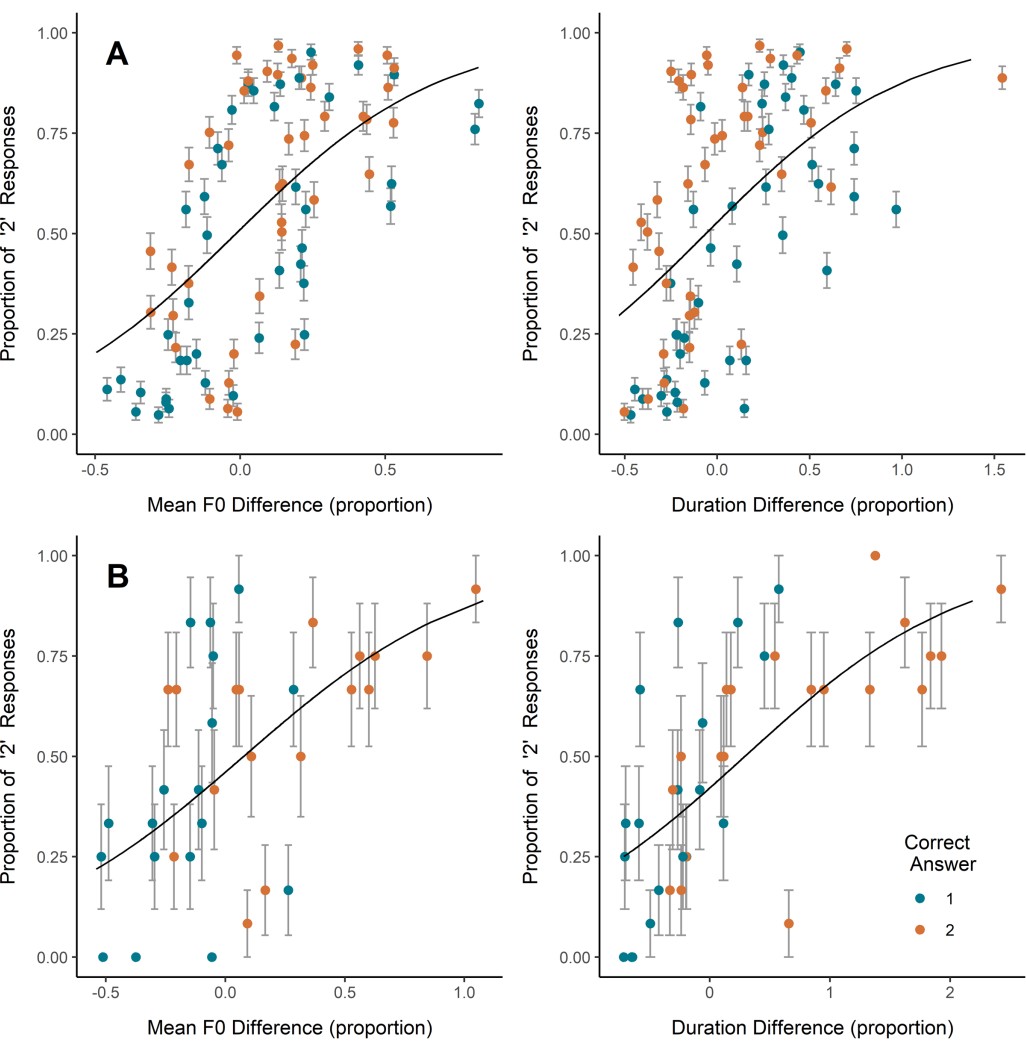

**Figure 2 Effects of F0 and duration on perceptions.** Relative increase or decrease from the first to the second vocalization in (A) coo trials (all participants) and (B) scream trials (experienced participants only since no significant effect was observed in inexperienced participants), plotted against the proportion of participants who indicated the second vocalization as reflecting a more intense emotion. Each point represents one stimulus pair; color represents whether the first or second vocalization in the pair represented greater arousal. Error bars represent SE. Trendlines reflect model predictions.

predictor to become nonsignificant and yielded a coefficient of correlation of fixed effects = −0.746, indicating collinearity. We therefore tested the correlation between relative differences in scream F0 and duration across stimulus pairs, and found a significant positive correlation (scream pairs with a larger difference in F0 tended to have a larger difference in duration; Pearson's r = 0.78, t = 7.18, $p < 0.001$), suggesting that the effects of F0 and duration on scream perception here are best interpreted together. As with responses to coos, each of these effects of acoustic differences on responses to scream stimuli was in the direction consistent with the mean relative acoustic difference between high- and low-arousal screams (Table 2). Examining Fig. 2B, it appears that the tendency shown by experienced participants to select the higher-F0 and/or longer scream led them

to respond accurately more often than not (*i.e.*, more green/dark points in the lower-left corner and more orange/light points in the upper-right corner).

## DISCUSSION

The hypothesis of evolutionary homology in emotional vocal communication predicts that humans should be able to accurately perceive the emotional significance of heterospecific vocalizations—specifically, to discern this based on acoustic variation within (not only between) call types. We tested this prediction by having participants listen to rhesus macaque coos and screams, and choose the vocalization they perceived to reflect greater emotional intensity. Inexperienced and experienced participants alike exhibited above-chance accuracy, relative to differences in emotional arousal between coos, as measured by increases in stress hormone levels, and between screams, as inferred from differences in the intensity of aggression received by vocalizers. Overall, these findings support the hypothesis that essential aspects of emotional vocal communication are shared between humans and rhesus macaques.

### Accuracy and experience

Overall accuracy was modest (mean accuracy less than 60%), though comparable with the findings of similar studies (*Filippi et al., 2017a*, *2017b*). That said, it is important to note the fine-grained nature of the discriminations we asked participants to perform. Whereas many similar studies have sampled vocalizations from relatively disparate contexts, *e.g.*, feeding *vs* aggression (*Faragó et al., 2017*) or isolation *vs* reunion *vs* nursing *vs* veterinary procedures (*Maruščáková et al., 2015*), our coos were all sampled from an identical controlled experimental behavioral test; the only factors differentiating the contexts from which high- and low-arousal coos were sampled were the characteristics and background of an individual macaque, shaping its subjective experience of the test, as reflected in stress hormone levels. Insofar as separation from the mother and the rest of the group (albeit for only 30 min) probably evoked a high-arousal, negative emotional state in every 6-month-old macaque, the emotional difference between coos within a pair was probably relatively small. The same reasoning applies to screams; almost every rhesus macaque scream occurs in a context associated with a relatively high-arousal, negative emotional state. Thus, while we operationally labelled vocalizations "high- and low-arousal," these labels are relative; it might in fact be more accurate to think of the arousal levels represented among our stimuli as ranging from "moderately high" to "very high." In light of that, our participants' accuracy was significantly above chance not only in a statistical but also in a biological sense. One explanation for these findings is that participants performed an anthropomorphic extrapolation from human emotional communication to that of rhesus macaques, perceiving emotion from vocalizations as though they were produced by humans. Such anthropomorphism could have resulted in accurate responses more often than the reverse, thanks in part to any similarities in the acoustic indicators of arousal stemming from humans' and macaques' shared evolutionary ancestry (discussed further below). Participants could also have drawn on experience with the communication of familiar nonhuman species, though surprisingly, having formally worked with non-

primate animals was associated with significantly decreased accuracy on coo stimuli. Participant characteristics other than experience—empathy, human emotion recognition, and animal attitudes—failed to show any relationship with responses, in line with one previous study (*Maruščáková et al., 2015*).

While even inexperienced participants performed with above-chance accuracy, we did find some effects of experience with rhesus macaques, supporting the hypothesis that familiarity with a species can improve sensitivity to emotional communication by that species (*Nicastro & Owren, 2003*; *Tallet et al., 2010*; *Parsons et al., 2019*). Experienced participants exhibited slightly (but statistically significantly) greater accuracy with scream stimulus pairs, and far more often chose the scream over the coo in mixed stimulus pairs, though experience had no effect on accuracy with coo stimulus pairs. One possibility is that humans might typically have more experience perceiving cues to emotion in vocalizations exhibiting the F0 of coos (generally 400–3,000 Hz) than that of rhesus macaque screams (generally 2,000–10,000 Hz), since the former is closer to the human vocal range. Previous research suggests listeners of a given species may be more sensitive to emotional vocalizations with a F0 falling within that species' typical range (*Root-Gutteridge et al., 2021*). Emotion perception from screams might therefore have been less intuitive for inexperienced human listeners, making experience a more significant factor. That said, tonal screams comprise only one of several rhesus macaque scream classes, warranting caution in extrapolating to screams more generally.

Interpreting the effect of experience on responses to mixed stimuli (containing one coo and one scream)—in which experienced participants much more often indicated the scream as reflecting a more intense emotion, while inexperienced participants did not—is more straightforward: several participants reported prior knowledge of the socioecological contexts associated with coos and screams, and indicated that they drew on this knowledge to judge screams as more emotionally intense than coos. Inexperienced participants, in contrast, presumably lacked such a basis of prior knowledge on which to form a consistent explicit strategy, and therefore responded in a much less consistent manner. Similarly, human listeners in a recent study did not provide significantly different ratings of emotional valence (positive *vs* negative states) to rhesus macaque feeding calls *vs* screams (*Fritz et al., 2018*). The effect of experience on responses to mixed stimuli was by far the strongest effect of any participant characteristic; its effects on accuracy with scream pairs was modest by comparison.

Experienced participants' knowledge of the contexts in which macaque screams occur might also have played a role in their greater accuracy on scream pairs (judging which of two screams was associated with greater arousal). The acoustic structure of screams is generally conserved across a phylogenetically diverse array of taxa, yet unlike nonhuman primates which produce screams primarily during aggression, human screams occur in a wide variety of emotional contexts, including positive ones, *e.g.*, intense pleasure-joy-elation, sadness, pain, anger-rage, and fear (*Engelberg, Schwartz & Gouzoules, 2021*; *Frühholz et al., 2021*). To speculate, these species differences could have influenced participants' responses, if perception of the macaque screams drew on processes similar to those involved in conspecific scream perception. Experienced participants, most of whom

reported knowing that macaque screams occur primarily during aggression, might have avoided any resulting biases to a greater extent, thus responding more accurately on scream trials. The question of how heterospecific scream perception is influenced by humans' expanded scream usage seems an intriguing area for further study.

## Acoustic cues to emotional arousal

Vocal F0 is a well-established correlate of emotional arousal across mammals including humans, and humans have been shown to perceive higher-F0 vocalizations from a variety of species (including conspecific vocalizations) as more emotionally intense (*Scherer, 2003*; *Juslin & Laukka, 2003*; *Nicastro & Owren, 2003*; *Pongrácz et al., 2005*; *Sauter et al., 2010*; *Briefer, 2012*; *Faragó et al., 2014*, *2017*; *Maruščáková et al., 2015*; *Filippi et al., 2017a*; *Kelly et al., 2017*; *Schwartz & Gouzoules, 2019*). This trend is probably due in part to phylogenetically widespread mechanisms of arousal and vocal production: arousal increases tension in the vocal fold muscles, resulting in faster oscillation, *i.e.*, higher F0 (*Scherer, 1986*; *Titze, 1994*; *Riede, 2010*; *Briefer, 2012*). In the present study, inexperienced and experienced participants alike tended to indicate the higher-F0 coo in a pair as more emotionally intense. Experienced, but not inexperienced, participants appeared to base scream selections partly on F0 as well, perhaps contributing to their greater accuracy. These findings are consistent not only with the literature in general but also with the findings of a recent study on rhesus macaque vocal production, which demonstrated a positive correlation between F0 and arousal among coos as well as screams (*Schwartz, Sanchez & Gouzoules, 2022*). Thus, vocal F0 appears to be a significant indicator of emotional arousal in humans and rhesus macaques (and other mammalian species) alike, probably due to evolutionary homologies in the physiology and anatomy of vocal production. In the present study as in previous ones (*Kelly et al., 2017*), humans' sensitivity to F0 seems to have led participants to respond correctly in some cases and incorrectly in other cases, but the former appears to have been more common than the latter in the present study (Fig. 2).

Given the significant role of F0 in within-call-type discriminations, it is puzzling that inexperienced participants did not consistently indicate high-F0 screams as reflecting a more intense emotion than lower-F0 coos. Likewise, participants in a recent study did not perceive similar call types by rhesus macaques as differing in emotional valence (*Fritz et al., 2018*). Just as F0 and other acoustic parameters seem to correlate more strongly with arousal within a call type than between call types from a vocal production standpoint (*Schwartz, Engelberg & Gouzoules, 2020*), our participants appeared to rely on F0 as a cue to emotional intensity more for within- than between-type discrimination. This observation is consistent with the "identification-attribution model" of nonverbal vocal perception, which postulates that classification or identification of call types is cognitively and perhaps neurally distinct from the attribution of emotional states to vocalizations (*Spunt & Lieberman, 2012*; *Sperduti et al., 2014*; *Anikin, Bååth & Persson, 2018*).

In addition to F0, inexperienced and experienced participants both tended to select a coo when it was longer in duration or noisier (lower HNR) than the alternative, consistent with previous studies of human emotion perception (*Faragó et al., 2014*; *Filippi et al.,*

*2017b*; *Schwartz & Gouzoules, 2019*; *Anikin et al., 2021*; *Massenet et al., 2022*). A recent study suggested a negative correlation between arousal and HNR in coos (*Schwartz, Sanchez & Gouzoules, 2022*), and among coo stimulus pairs used in the present study, the higher-arousal coo was noisier on average (Table 2). Thus, the effects of noisiness on participants' decisions might thus have led them to respond correctly on some trials. It should be noted, however, that perception of noisiness is not fully distinct from that of F0; chaos and other nonlinear phenomena can affect perceived pitch (*Anikin et al., 2021*). Participants' tendency to select the longer coo likely played no role in their overall accuracy, instead leading them to the correct and incorrect responses an approximately equal proportion of the time (Fig. 2A). Experienced participants also tended to select the longer scream, though the correlation between scream F0 and duration made their effects difficult to disambiguate. Finally, inexperienced participants showed a preference for the coo or scream in a pair that showed a wider F0 range, though estimates of these effects were small.

Overall, consistencies between our findings and those of other studies suggest that human listeners might use similar acoustic cues to assess the emotional arousal represented by acoustic variants within a call type, whether it be a human nonverbal vocalization (*Sauter et al., 2010*; *Schwartz & Gouzoules, 2019*), a vocalization of a species with which the listener is familiar (*Nicastro & Owren, 2003*; *Faragó et al., 2014*, *2017*; *Massenet et al., 2022*), or a vocalization of an unfamiliar species (*Filippi et al., 2017a*; *Kelly et al., 2017*). These perceptual rules of thumb (*e.g.*, a higher-F0 sound indicates higher arousal) seem to lead to accurate perceptions more often than not, thanks in part to broad similarities in the acoustic indicators of arousal owing to shared evolutionary ancestry. However, the utility of these rules of thumb in making accurate judgments, and the role of familiarity and experience in determining the salience of different acoustic cues, might vary across species and call types.

## Emotional arousal and valence

A remaining issue is the distinction between emotional arousal and valence, *i.e.*, the spectrum from negative to positive states (*Russell, 2003*; *Mendl, Burman & Paul, 2010*; *Briefer, 2012*). Participants were not instructed on the distinction between emotional valence and arousal, and thus it is possible that some of them might have interpreted "a more intense emotion" to mean a state that is more negative as well as more aroused. However, the term "intensity" is often used interchangeably with "arousal," and previous research has shown that listeners provide similar ratings for both terms with respect to vocal stimuli (*Laukka, Juslin & Bresin, 2005*). In truth, the stimulus pairs might have represented differences in emotional valence in addition to arousal: higher concentrations of stress hormones might reflect a more negative state than lower concentrations, and contact aggression might evoke a more negative emotional reaction than non-contact aggression. As in many other studies of animal emotional vocal communication (reviewed in *Briefer (2012)*), the present one does not permit us to discern the precise roles of emotional arousal and valence. With that said, the mechanisms by which emotional arousal affects the voice are more straightforward than for valence (*Briefer, 2012*).

F0, a well-established correlate of emotional arousal but not valence, strongly influenced our participants' responses. For these reasons, it seems most parsimonious to conclude that emotional arousal played a role in our findings.

## CONCLUSIONS

The present study demonstrated that listeners with and without job-related experience with rhesus macaques have the capacity to discern fine-grained distinctions in the emotional arousal level represented in the coos and screams of that species. Experience was associated with limited improvement in this ability, but the most obvious effect of experience was seen in between-vocalization-type discriminations, where semantic knowledge about the socioecology of coos and screams appeared to influence responses. Thus our findings are consistent with the hypothesis that humans' capacity for accurate emotional perception of rhesus macaque vocalizations stems in part from the evolutionary ancestry shared by the two species. Specifically, it appears that listeners utilized acoustic cues that reliably indicate emotional arousal in humans and rhesus macaques alike due to homology in the physiology and anatomy of arousal and vocal production. Some nonhuman calls are directly evolutionarily homologous with human nonverbal vocalizations, screams being one example. A remaining question is how humans' usage of such vocalizations, as in the contextual expansion seen in human screams, influences perceptions of heterospecific calls. Future research examining emotion expression within a variety of call types of a phylogenetically diverse array of taxa, and human perception thereof, will further improve our understanding of the evolutionary history of emotional communication.

## ACKNOWLEDGEMENTS

We thank Drs. Mar M. Sanchez and Mark E. Wilson for the design and supervision of the human intruder and cortisol testing used to generate the coo stimuli. Thanks also to Natalie Brutto, Kelly Bailey, Manuel Bautista, Patrice Rando, Jalani Paul, Jonathan Engelberg, Jodi Godfrey, and Desiree De Leon for their assistance in collection of those human intruder and cortisol data and coo recordings, and Emma Satty for her assistance collecting participant responses. Finally, we thank Dr. Andrey Anikin and one anonymous reviewer for their feedback and suggestions for revisions to this manuscript.

### Funding

This study was supported by the U.S. National Science Foundation Graduate Research Fellowship under Grant No. DGE – 1343012, by the U.S. National Institutes of Health (NIH) Grant Number 1R01HD077623 and the NIH Office of the Director, Office of Research Infrastructure Programs, P51OD011132 (Emory National Primate Research Center-ENPRC-base grant). The ENPRC is fully accredited by AAALAC, International. The funders had no role in study design, data collection and analysis, decision to publish, or preparation of the manuscript.

## Grant Disclosures

The following grant information was disclosed by the authors:

U.S. National Science Foundation Graduate Research Fellowship: 1343012.

U.S. National Institutes of Health (NIH): 1R01HD077623.

NIH Office of the Director, Office of Research Infrastructure Programs, Emory National Primate Research Center (ENPRC): P51OD011132.

## Competing Interests

The authors declare that they have no competing interests.

## Author Contributions

- Jay W. Schwartz conceived and designed the experiments, performed the experiments, analyzed the data, prepared figures and/or tables, authored or reviewed drafts of the article, and approved the final draft.
- Harold Gouzoules conceived and designed the experiments, performed the experiments, authored or reviewed drafts of the article, and approved the final draft.

## Human Ethics

The following information was supplied relating to ethical approvals (*i.e.*, approving body and any reference numbers):

Human testing procedures were approved by and conducted in compliance with the Emory University Institutional Review Board.

## Animal Ethics

The following information was supplied relating to ethical approvals (*i.e.*, approving body and any reference numbers):

All animal procedures were approved by the Emory University Institutional Animal Care and Use Committee in accordance with the Animal Welfare Act and the U.S. Department of Health and Human Services "Guide for Care and Use of Laboratory Animals."

## Data Availability

The raw data are available in the Supplemental Files.

## Supplemental Information

Supplemental information for this article can be found online at http://dx.doi.org/10.7717/peerj.14471#supplemental-information.

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
