# Peer review of "Humans read emotional arousal in monkey vocalizations: evidence for evolutionary continuities in communication"

_PeerJ, doi:10.7717/peerj.14471_

## Round 0.1 · original submission · Minor Revisions

I have now received the reviewers' comments on your manuscript. They have suggested some minor revisions to your manuscript. Therefore, I invite you to respond to the reviewers' comments and revise your manuscript.

·

Basic reporting

OK

Experimental design

OK

Validity of the findings

OK

Additional comments

Schwartz and Gouzoules present a study in which human listeners judged the level of arousal in two vocalization types by rhesus macaques, both of which are known to vary acoustically as a function of the animal’s arousal level. The sample size was 112 listeners in the “inexperienced” group and 12 in the “experienced” group. The change in blood cortisol levels after the event eliciting coo vocalizations served as a physiological measure of stress, while screams were recorded in natural social encounters. High/low-arousal coo pairs were composed of calls from different individuals (why?), whereas scream pairs were taken from the same animal and were defined as high-arousal if there was “agonistic physical contact”. The authors report that listeners were (barely) above chance when guessing which of two stimuli was associated with higher arousal, and having previously worked with macaques improved the accuracy in the case of mixed coo-scream pairs.

The study seems to be well designed and carefully presented. I find it particularly impressive that there is an objective, physiological measure of arousal for each coo vocalization. My comments below are fairly minor and would require only moderate revisions to address.

Abstract “our findings ... generate hypotheses about how human nonverbal communication influences perceptions of other species' vocalizations” What kinds of hypotheses? This seems a bit vague.

Introduction Phylogenetic proximity vs familiarity vs domestication: a potentially relevant paper is Maigrot 2022 https://bmcbiol.biomedcentral.com/articles/10.1186/s12915-022-01311-5

Results
The key result, namely listeners being above chance when guessing which stimulus in a pair corresponds to higher arousal, is supported by t-tests on aggregated data, even though the rest of analyses correctly use the much more powerful binomial GLMM. Why not use unaggregated data and GLMM throughout?

The acoustic correlates of arousal are summarized briefly and informally as mean ± SD of differences between pairs of stimuli in Table 2. However, you have valuable physiological data – changes in blood cortisol, a continuous variable that can be used to predict coo duration, fo, etc. For screams, there are only two “objective” arousal values, high or low, but it would still be interesting to compare the acoustics of high/low-arousal screams more fully and formally. In addition, I would turn the question around and predict the “true” arousal level from acoustics because the classification accuracy of these models (separately for coos and screams) would provide a useful benchmark against which to interpret the low classification accuracy by human listeners.

line 456 “inclusion of one effect reduced the estimated size of the other, indicating collinearity” This is strange: collinearity is usually defined as excessively high correlation between two predictors that inflates standard errors, not effect sizes – if anything, effect sizes are expected to change in multiple regression due to masking etc., that’s the whole point of using this method. In general, I wonder if collinearity is really such a problem in this study. For example, are fo and duration really correlated so strongly that they cannot be used simultaneously as predictors (e.g., r > 0.8-0.9)? If so, it may be worth reporting these correlations.

Fig. 3 I suspect (correct me if I’m wrong!) that the regression lines are produced with geom_smooth() on just the aggregated proportions per pair rather than taken from the GLMM. This is misleading as it ignore the SEs around these points and thus radically downplays the uncertainty, which is particularly problematic for panel B.

Reviewed by Andrey Anikin

Reviewer 2 ·

Basic reporting

The title of the article is excellent and attractive. All the contents in different sections are specified in detail and completely systematically, and for this, I congratulate the respected authors. But there are points that, if revised, will show flawless work.
Rewrite the abstract in a structured way and mention the necessary items in each part.
The introduction section is long, I read it several times but I couldn't find a paragraph that didn't need to be said. Dear author, please explain a little more briefly, considering the nobility of the whole article.
In the method section, mention the type of study, sampling method, how to calculate the sample size and the reason for determining the number in each group.

Experimental design

no comment

Validity of the findings

no comment

Additional comments

Of course, one point is also important. The reason is that your research showed that humans have cognitive systems that are able to recognize emotions in non-verbal communication of primates, and you consider this a confirmation of Darwin's theory of evolution. But can't the similarity of the perceptual structure of humans to anthropomorphic animals such as primates be interpreted in another way? That humans are created with this intelligent and advanced structure that allows them to perceive non-human species. With all the respect and value I attach to your valuable research, in my opinion, this study and its results cannot be considered a confirmation of Darwin's theory. One can simply infer the structural similarity between perceptual structures between humans and non-humans.

---

## Round 0.2 · accepted · Accept

In my opinion, this manuscript has been revised with attention to the reviewers' comments and can now be published.

·

Basic reporting

OK

Experimental design

OK

Validity of the findings

OK

Additional comments

I am grateful to the authors for carefully addressing my comments and can recommend it for publication.

Just as a little aside:
“We elected to use one-sample t-tests over GLMM because we don’t think that GLMM can compare a distribution of scores to a predicted mean value” The significance of the intercept in a model like correct ~ 1 + (random effects) tests the hypothesis that intercept = 0 on the log-odds scale, which corresponds to 50%. Thus, the p-value you need is present in model summary – very simple.

Also, if duration and f0 correlate with r = .8 and both predict higher arousal ratings, I find it totally legitimate to put both simultaneously in multiple regression to see which one is actually driving the effect. In the revised text you report that “including both predictors in a single model caused one predictor to become nonsignificant” – it’s quite likely that the one that remains significant (f0, I imagine?) is the one that listeners attend to, and the correlation of arousal with duration is basically spurious. This kind of analysis is the raison-d’etre of multiple regression, and collinearity is only a problem if the correlation between predictors is too high relative to the amount of available data. Please forgive me for being so pedantic about the stats!